# Air Pollution inside Vehicles: Making a Bad Situation Worse

**DOI:** 10.3390/ijerph20216970

**Published:** 2023-10-25

**Authors:** Naowarut Charoenca, Stephen L. Hamann, Nipapun Kungskulniti, Nopchanok Sangchai, Ratchayaporn Osot, Vijj Kasemsup, Suwanna Ruangkanchanasetr, Passara Jongkhajornpong

**Affiliations:** 1Faculty of Public Health, Mahidol University, Bangkok 10400, Thailand; naowarut.cha@mahidol.ac.th; 2Thailand Health Promotion Institute, Bangkok 10330, Thailand; 3Tobacco Control Research and Knowledge Management Center, Bangkok 10400, Thailand; stephen.ham@mahidol.ac.th (S.L.H.); vijj.kas@mahidol.ac.th (V.K.); suwanna.rua@mahidol.ac.th (S.R.); passara.jon@mahidol.ac.th (P.J.); 4Faculty of Business Administration, Bangkok-Thonburi University, Bangkok 10170, Thailand; nopchanok.san@bkkthon.ac.th (N.S.); ratchayaporn.oso@bkkthon.ac.th (R.O.); 5Department of Ophthalmology, Faculty of Medicine, Ramathibodi Hospital, Mahidol University, Bangkok 10400, Thailand

**Keywords:** smoking, vehicles, PM_2.5_, air pollution, restrictions, children, Thailand

## Abstract

Thailand has successfully forwarded Article 8, Protection from Exposure to Tobacco Smoke, of the World Health Organization’s Framework Convention on Tobacco Control (WHO FCTC). It achieved its 100% smoke-free goals in public places in 2010, next pursuing other bans in outdoor places to lower particulate matter air pollution (PM_2.5_). Our aim was to expose the secondhand smoke levels in vehicles since SHS is a danger to everyone, but especially to children and youth. This is the first experimental study of its kind in Thailand. We measured PM_2.5_ for 20 min under four conditions in 10 typical Thai vehicles, including commonly used sedans and small pickup trucks. We used an established protocol with two real-time air monitoring instruments to record PM_2.5_ increases with different vehicle air exchange and air conditioning conditions. Monitoring was recorded in the vehicle’s front and back seats. The most common Thai ventilation condition is all windows closed with fan/air conditioning (AC) in operation because of Thai tropical conditions. Mean exposure levels were three and nearly five times (49 and 72 μg/m^3^) the 24 h WHO standard of 15 μg/m^3^ in the back and front seats, respectively. These high PM_2.5_ exposure levels warrant action to limit vehicle smoking for public health protection.

## 1. Introduction

While smoking is a top burden of disease risk factor, secondhand smoke exposure is also cited as a major burden of disease risk factor as a separate element of air pollution. 

The health consequences causally linked to smoking include cancers of the oropharynx, larynx, esophagus, trachea, bronchus and lung, stomach, liver, pancreas, kidney and ureter, cervix, bladder, and colorectal area, as well as acute myeloid leukemia.

Other chronic diseases include stroke, blindness, cataracts, age-related macular degeneration, congenital defects from maternal smoking (orofacial clefts), periodontitis, aortic aneurysm, early abdominal aortic atherosclerosis in young adults, coronary heart disease, pneumonia, atherosclerotic peripheral vascular disease, chronic obstructive pulmonary disease, tuberculosis, asthma, other respiratory effects, diabetes, reproductive effects in women (including reduced fertility), hip fractures, ectopic pregnancy, male sexual dysfunction (erectile dysfunction), rheumatoid arthritis, and immune function. 

The health consequences causally linked to exposure to secondhand smoke include the following:

In children: middle ear disease, respiratory symptoms, impaired lung function, lower respiratory illness, and sudden infant death syndrome.

In adults: stroke, nasal irritation, lung cancer, coronary heart disease, and reproductive effects in women (low birth weight) [1]. 

There is a synergistic effect with smoking and exposure to secondhand smoke since different solid and vapor phase constituents are involved in each and evidence shows that the combination of both is detrimental beyond each separately [2]. PM_2.5_ levels are part of this synergy, but particulate levels from smoking and secondhand smoke are usually assessed together as additive. This means that despite PM_2.5_ levels representing general air pollution levels, they are not the only exposure type in air pollution, as the World Health Organization indicates in listing air pollution standards for multiple pollutants shown in Table 1 [3]. 

Smoke-free policies have been essential tobacco control measures for more than 50 years. These policies initially focused on banning smoking in confined spaces, such as in airplanes. Even with this emphasis, it took many years for smoking in planes to be completely banned [4]. Public conveyances were some of the first places smoking was banned. However, smoking in private vehicles did not receive initial attention, as smoke-free public places were the focus and became widely adopted due to the World Health Organization Framework Convention on Tobacco Control (FCTC), which took effect in 2005 with a time-limited goal of complete smoke-free public places in five years by countries who adopted the FCTC treaty [5].

By 2007, 146 countries had ratified the FCTC, and Article 8, Guidelines for Protection from Tobacco Smoke Exposure, had been adopted at the Second Conference of the Parties (COP 2) held in Bangkok, Thailand. Early research on smoking in cars began in 2006 [6,7]. Over thirty-five studies were completed between 2010 and 2015, and the first systematic review and meta-analysis of initial smoke exposure levels and restrictions on smoking in cars was completed in 2015 [8]. From 2016 to the present, research on secondhand smoke (SHS) in vehicles decreased, although countries adopted many smoke-free laws in public places. The WHO Report on the Tobacco Epidemic 2021 indicated that 34.5 percent of countries completely met Article 8 guidelines on protection from tobacco smoke [9]. Full and partial compliance did not reach half of the 194 FCTC countries, and this did not include restrictions on private vehicle smoking. Only 12 countries have laws restricting smoking in vehicles with children [10]. This lack of attainment of FCTC smoke-free goals is significant and likely due to the substantial interference by the tobacco industry in delaying air pollution and secondhand smoke legislation [11].

In Thailand, local research on SHS in public places started in 2002 and invested in demonstrating the levels of PM_2.5_ in locations where smoking was still allowed. The research was conducted in restaurants, public buildings, transport terminals, pubs, bars, and airports. After 2005, the Thai Government began expanding the areas covered by smoke-free restrictions through notifications of the Nonsmokers’ Health Protection Act of 1993. Ten notifications in 2007 moved many public places from allowing smoking areas to banning smoking completely [12]. By 2010, Thailand attained complete smoke-free coverage of public areas, including airports, bars and pubs, educational facilities, healthcare facilities, hotels, public transportation, restaurants, shops/shopping complexes, transport terminals, universities, and workplaces/offices [13]. While Thailand did not ban smoking in cars, it did follow its bans in indoor public places with outdoor bans in parks, markets, and beaches with the 2017 Tobacco Product Control Act [14]. Our study of PM_2.5_ exposures in cars in Thailand focuses on explicitly addressing the pollution levels generated in typical vehicles used in Thailand with air exchange and ventilation conditions commonly employed in vehicles in Thailand’s tropical climate. We aim to expose the secondhand smoke levels in vehicles since SHS is a danger to everyone, but especially to children and youth.

## 2. Materials and Methods

Secondhand smoke exposure was assessed inside cars by measuring PM_2.5_ under various experimental conditions. PM_2.5_ is recognized as a standard indicator of particulate pollution that produces symptoms and can lead to disease and death [15]. Ten car owners smoked one or more cigarettes in their cars as they usually do intermittently under the conditions of this experiment. Only one car owner smoked more than one cigarette during all 20 min air exchange/conditioning experimental sessions. All smokers completed four controlled SHS exposure conditions beginning with the condition of least exposure with open window air exchange and mechanical air conditioning. This study followed steps in previous research to assess secondhand smoke exposure levels [16]. The study was reviewed and approved for human subject participation by the Ethics Committee of the Thailand Health Promotion Institute. 

Individuals who smoked and owned cars were recruited by colleagues who knew of our research project in an area on the outskirts of Bangkok in 2020 and 2021. See Figure 1 and Figure 2 of a car and truck in this study, respectively. Car owners completed a screening questionnaire to identify themselves as smokers, that they owned a typical vehicle, and that they regularly smoked while driving. All participants regularly smoked in their cars and consented to participate in the four experimental conditions of this study. The air exchange design and mechanical air ventilation and conditioning in this study are typical for vehicles in a tropical setting.

The four conditions assessed included:

Condition 1. Participants smoked inside their vehicle with all windows open during a 20 min drive (with ventilation and air conditioning in operation).

Condition 2. Participants smoked inside their parked vehicle with all windows open, the engine running, ventilation, and air conditioning in operation.

Condition 3. Participants smoked inside their vehicles with all windows completely closed except the driver’s window, which was halfway down, during a 20 min drive with ventilation and air conditioning in operation.

Condition 4. Participants smoked inside their parked vehicles with all windows closed except the driver’s window, which was halfway down, with the engine running, ventilation, and air conditioning in operation. 

The climate-control fan inside the car and the air conditioning were set at the intermediate setting in all conditions. The car doors and windows were opened for at least 5 min between each experimental condition to remove some secondhand smoke. See Figure 3 to see how vehicles were opened between experimental conditions. Since research shows that it can take up to one hour for secondhand smoke to clear significantly from a vehicle, there was a cascading increase in the base level of secondhand smoke at the beginning of each succeeding condition [17].

The sequence of conditions for each participant did not vary, since they were arranged from the lowest to highest exposure level based on the airflow/ventilation level of the conditions being tested. 

Air quality monitoring equipment was used to measure the level of PM_2.5_ for 20 min in the car during each condition and for at least 5 min in the car before the next condition, with the starting level recorded before each condition so the increase from smoking in the car could be determined. “Air quality in each vehicle was monitored using two TSI Sidepak aerosol monitor” [18]. “The Sidepak was used with a 2.5-micron impactor to measure PM_2.5_ and was calibrated prior to each experimental session with a high-efficiency particulate air filter according to the manufacturer’s specifications. The Sidepak was set to record the average PM_2.5_ concentration every 60 s. A customized calibration factor of 0.32 was applied to the devices, determined by calibrating the devices in the present study with other light-scattering photometers measuring tobacco smoke particulates” [6,19]. A Bangkok Sidepak calibration center calibrated the instruments used in this investigation [20]. Monitoring was conducted on one car at a time.

Two monitors were secured in the vehicle being tested. One monitor was located in the front passenger’s seat and one in the middle of the back seat of each car so that the data collected would provide a reasonable estimate of exposure levels of PM_2.5_ for a young child sitting in the car’s back seat. See Figure 4 of the placement of monitoring instruments in the front and back seats of each vehicle.

Once the equipment was secured, the participant received specific instructions about the condition being monitored. Before the experimental condition was tested, “the participant was instructed not to turn on the car, open any windows or doors while inside the car, or turn on the air conditioning or fan unless specified for testing the condition. 

Once in the car, the participant lit the cigarette and smoked it naturally while under the experimental condition being evaluated” [6]. The participant either finished the cigarette and immediately left the vehicle (test of the stationary vehicle) or drove for 20 min while smoking his cigarette(s) before returning and exiting the vehicle. In all cases, the time from the door opening to the door shutting again during the exit period was less than 5 s and did not appear to affect the levels of PM_2.5_ in the car. 

The participant could smoke one or more cigarettes during each condition, which was documented. The air monitoring device remained in the car for at least 25 min during each condition to provide baseline comparison values before the car started and after smoking occurred. In Conditions 1 and 3 where participants drove their cars, the participant was asked to remain on roadways and streets in the residential area, maintaining speeds below 50 km/h while obeying local traffic signs and regulations. Data were collected from monitors in four-door sedans, four-door pickup trucks, and one two-door pickup with one monitor only since there was no rear seat. These are the most common passenger vehicles in Thailand. According to the manufacturer’s specifications, the vehicles’ average size of the interior cabin space was 2.6 m^3^, ranging from 2.4 to 2.9 m^3^. All participants smoked their regular tobacco cigarette brand during the four experimental conditions.

TrakPro software, version 3.41, was used to download data from the TSI Sidepak for analysis [21]. Data were then exported to a spreadsheet program. The Sidepak recorded PM_2.5_ levels every minute that were averaged before and during each vehicle testing in all conditions. 

## 3. Results

Table 1 and Table 2 show findings from ten vehicles: five sedans, four large (four-door) pickup trucks, and one smaller pickup truck (two-doors). Pickup trucks were chosen since they are popular passenger vehicles in Thailand, making up 45.7% of all vehicles sold in Thailand in 2021 [22]. As described in the methodology, four conditions for each vehicle were monitored from the open condition while moving to the driver window half down stationary condition. In all conditions, the fan and air conditioning of the vehicle were operated since this is the most common type of air exchange and ventilation/AC in vehicles in Thailand since hot temperatures are present year-round during daytime hours.

The particle matter dispersed in a vehicle depends on the vehicle’s internal space. The internal space varied only slightly, in descending order for sedans, four-door trucks, and one two-door truck. We proceeded through all four conditions in each vehicle with only a 5 min airing of each vehicle’s internal space. We measured base starting levels of PM_2.5_ in the vehicle at the beginning of monitoring levels for each condition so the increase in PM_2.5_ could be determined due to smoking in the vehicle. Table 2 and Table 3 below summarize the average mean level PM_2.5_ for the front and back seats (only front seat for the 2-door truck) in the ten vehicles monitored in the four conditions of this study.

All increases are beyond the WHO 24 h PM_2.5_ standard and nearly beyond the interim Thai standard of 37.5 in all cases as well. The interim standard is set for countries to move lower to the WHO’s 15 μg/m^3^. 

All conditions of air exchange and mechanical air ventilation/condition have PM_2.5_ levels above the WHO 24 h PM_2.5_ standard, and three conditions (one in the front seat and two in the back seat) are above the Thailand 24 h standard, an interim PM_2.5_ established as a temporary standard in reducing the PM_2.5_ level to the ideal WHO standard.

Results in the back seats where children would probably be seated had lower PM_2.5_ levels than in the front seats, but the percent increase above the base level of each condition for each vehicle was nearly as high as that for increases in the front seat. For some conditions, the percent increase in the back seat reached 70% of the level in the front seat. Note that the most common condition in Thailand is condition 3, and all condition 3 increases in the front and back seats were above the 24 h WHO and Thai standards. In addition, in common conditions 1 and 3, peak exposure levels not reported in the Tables rose an average of 56 and 180 μg/m^3^ in the front seat and 29 and 75 μg/m^3^ in the back seat. These are 12 and 5 times the 24 h PM_2.5_ standard, respectively.

## 4. Discussion

Our findings show that PM_2.5_ mean levels in the front seat increased by 13–149 μg/m^3^ inside vehicles exclusive of the outdoor base PM_2.5_ where the smoking of one cigarette was present (Table 2), and levels in the back seats increased by 5–107 μg/m^3^ from base levels (Table 3), reaching between 40 and 70% of front seat levels depending on condition. The results are very high in condition 3 of vehicle air exchange/ventilation, the most common condition for drivers in Thailand. The mean outdoor ambient air level was 17.1, slightly less than the last reported Thai average of 20 μg/m^3^ in 2021 [23]. This level is already above the 24 h WHO standard for PM_2.5_. Thailand raised its interim annual standard to 15 μg/m^3^ and its 24 h standard to 37.5 μg/m^3^ in 2023. The mean PM_2.5_ level of experimental condition 3 combined with the Thai outdoor level in the back seat is more than three times the WHO 24 h standard and above Thailand’s interim standard (49 > 37.5). These findings are consistent with several other studies showing significant increases in PM_2.5_ from car smoking. For example, Sendzik and colleagues concluded, “In moderate ventilation conditions (air conditioning or having the smoking driver hold the cigarette next to a half-open window), the average levels of PM_2.5_ were reduced but still at significantly high levels (air conditioning = 844 μg/m^3^; holding cigarette next to a half-open window = 223 μg/m^3^” [16]. Semple also investigated PM_2.5_ levels but during typical travels in passenger vehicles. He concludes, “During smoking journeys, peak PM_2.5_ concentrations averaged 385 μg/m^3^), with one journey measuring over 880 μg/m^3^. PM_2.5_ concentrations were strongly linked to the smoking rate (cigarettes per minute). Use of forced ventilation and opening of car windows were widespread during smoking journeys, but PM_2.5_ concentrations were still found to exceed WHO indoor air quality guidance…” [24]. Our results were extreme when vehicle drivers smoked more than one cigarette in the typical condition of driving with the window halfway down and operating with moderate air exchange and air conditioning. With the driver smoking two cigarettes, the front and back seat increases were 256 and 176 μg/m^3^, respectively. With three cigarettes smoked, the front and back seat increases were 549 and 432 μg/m^3^, respectively. Multiple other studies confirm and discuss that air pollutants from car smoking rise above standards by the US EPA and WHO and produce several dangerous peak air pollution levels [25,26,27,28,29,30,31,32]. These levels are extreme enough to cause acute symptoms in children and, therefore, should prompt immediate steps to establish restrictions on smoking in cars with child/adolescent passengers. 

It is essential to highlight the fact that research is bringing attention to the need for restrictions on smoking in vehicles. Systematic reviews of research on smoking in cars show the highly toxic pollutants produced and that removing such exposures has positive health consequences [8,31]. In 2015, a systematic review of atmospheric and biological markers of SHS exposure in cars, with twelve studies meeting inclusion criteria, showed that PM_2.5_ ranged widely depending on vehicle natural and mechanical ventilation. Factors that affected levels included air-conditioning status, the extent of airflow, and driving speed. Because extremely high exposures occurred even with air-conditioning and high airflow ventilation, the authors conclude that eliminating smoking in cars is necessary for containing air pollution levels in line with air pollution standards for short-term exposure.

“In a 2021 meta-analysis with ten effect estimates from four studies, smoke-free car policies were associated with an immediate tobacco smoke exposure (TSE) reduction in cars (risk ratio 0.69, 95% CI 0.55–0.87; 191,466 participants) with heterogeneity of results substantial (80.7%; *p* < 0.0001). One additional study reported a gradual TSE decrease in cars annually. Individual studies found TSE reductions on school grounds, following a smoke-free school policy, and in-hospital attendance for respiratory tract infection, following a comprehensive smoke-free policy” [31]. In places such as Canada, England, Italy, Maine in the US, and Scotland, where follow-up studies were conducted on air pollution levels after restrictions on smoking in cars with children, overall PM_2.5_ levels have decreased, notably both from the compliance with the law and the more focused awareness these restrictions bring to PM_2.5_ air pollution levels [30,32,33,34,35].

There is also survey evidence that the public supports restrictions on smoking in cars. A 2023 study that looked at attitudes to smoke-free policies showed high support for smoke-free cars carrying children (86%, 95% CI: 81–89) and in playgrounds (80%, 95% CI: 74–86) and school grounds (76%, 95% CI: 69–83) [36]. In Thailand, a recent newspaper editorial on air pollution titled “Time to Take Axe to PM_2.5_” concludes, “The goal is as clear as the air we need and deserve—PM_2.5_ must be lowered” [37].

While it is known that PM_2.5_ air pollution exposure increases all-cause mortality by 3–26%, the episodic nature of car exposure has been used to question the need for a smoke-free policy in cars. However, several studies have shown that instituting smoke-free car legislation not only addresses the acute danger to vulnerable populations like children, pregnant women, and the elderly but also reduces overall secondhand smoke exposure beyond existing smoke-free policies [38]. This finding, in combination with the continuing concern about youth exposure to secondhand smoke in the home, highlights the need to take action on youth tobacco-related exposures. In 2019, Thailand passed the Family Development and Protection Act, holding to account those who smoke in the home for the toxic exposure of others to tobacco smoke. Those who violate the law can be tried in juvenile or criminal court. Recent research shows how smoking in the home often results in higher air pollution levels than supposed [39]. In the past, smoking in a private car was not understood to cause harm. Today, various actions inside a vehicle are prohibited by law because they can cause public harm. Smoking in hired vehicle transportation (taxis) is already banned in Thailand. Not using a seat belt or putting a child in a car seat is illegal in many countries. Engaging in behaviors that distract drivers, like texting while driving, are examples of actions in one’s car that are now illegal.

Removing actions that normalize dangerous behaviors like nicotine addiction is a concern of parents who want their children to remain free from immediate and long-term health consequences. In addition to the distractions that can result from smoking in cars, it is not lost on health officials that modeling smoking behavior in the home and car significantly influences children and youth. Laverty, citing recent research, states that smoke-free car policies “are justified by both trying to denormalize tobacco smoking near children and by the very high air concentration of toxins upon smoking within cars” [38]. In a later comment on recent Scottish findings, he notes that smoke-free car legislation can produce additional health benefits over and above existing smoke-free laws, and multiple environmental benefits provide the impetus to accelerate the implementation of comprehensive smoke-free car policies across the globe [38]. 

Because of growing air pollution levels from fires and other sources, the reduction in PM_2.5_ is becoming recognized as critical to sustaining life. Research shows that every 10 μg/m^3^ increase in ambient PM_2.5_ increases all-cause mortality between 3 and 26%, chances of childhood asthma by 16%, lung cancer by 36%, and heart attacks by 44%. [40]. The World Health Organization estimates 6.7 million indoor and outdoor air pollution deaths yearly. They emphasize that “By reducing air pollution levels, countries can reduce the burden of disease from stroke, heart disease, lung cancer, and both chronic and acute respiratory diseases, including asthma” [41]. Health burdens from air pollution by region are most extreme in the WHO Southeast Asia and Western Pacific Regions, which includes Thailand [42].

It is recognized that the pollutant that is responsible for most air pollution deaths is particulate matter. Air pollution deaths can be from acute or longer-term exposures, which also generate disabling chronic disease conditions. Multiple risk factors can play a role in air pollution deaths. For example, lung cancer can result from the separate and synergistic effects of poor ambient air quality and cigarette smoking [43]. Also, exposure to air pollution from smoking and smoke has epigenetic consequences, with later disease from earlier exposures [44]. 

Given the evidence of dangers from PM_2.5_, what can be done to protect from secondhand smoke in cars? Monitoring PM_2.5_ in areas where exposures are dangerous to health is a concern beginning to be addressed in Northern Thailand. However, it is vital to address both high ambient air levels in Thailand and the indoor threat that secondhand smoke poses to children, pregnant women, and seniors. Public health practice suggests that action is needed in all sectors of society and with protections across all ages [45].

### Limitations of This Study

A primary limitation of this study is the constraints placed on it by the COVID-19 epidemic, which caused limited time and opportunity for testing exposures in vehicles. We had hoped to test more vehicles of various types, with various interior cabin spaces, varied levels of air exchange and air conditioning, and increased numbers of cigarettes smoked in each experimental condition. Because our research time period was limited and COVID-19 made it impossible to get vehicles of various types with different interior cabin spaces, and we were not able to test the number of vehicles needed to compare different air exchange and ventilation conditions or a range of cigarettes smoked, we concentrated on measuring PM_2.5_ inside vehicle cabins of typical Thai vehicles with a relatively low smoking rate (1 cigarette in 20 min) and standard air exchange and ventilation settings. Our inability to overcome limitations in testing varying factors did not keep us from showing high levels above the WHO 24-h standard in typical conditions.

Another limitation of our findings is the concern that providing evidence of the high levels of PM_2.5_ may not translate into concern for the resulting consequences. Despite finding that the most common air exchange and conditioning vehicle practice (condition 3) results in three (backseat) to five (front seat) times the 24 h WHO standard, past studies may make our results seem low. Semple, in discussing the mixed results from various studies, notes that “some studies only measure concentrations during the time when smoking took place, whereas our data relate to typical real-life journeys, which involved a mixture of non-smoking and smoking time periods” [24]. He also cites Ott et al. who notes “that ventilation, air conditioning, window position and car speed all influenced SHS concentrations in cars. Such factors combined with different participant smoking behaviors such as differences in study design may account for the variability in PM_2.5_ concentration across studies” [46]. Our study used a naturalistic approach for the number of cigarettes smoked in the monitoring period and included a variety of ventilation conditions to see how these conditions affected PM_2.5_ results. The mixed design makes the comparison of our significant results with other studies difficult.

Even with the past before-and-after results of restrictions on smoking in cars, it is difficult to get policymakers to accept that secondhand smoke in cars is a significant problem. It is insufficient to point out that breathing is vital and that high levels of little things like particulates can kill. Additional studies with different metrics that compare and visualize the levels of PM_2.5_ inside standard Thai vehicles are needed to reinforce our findings of PM_2.5_ levels that affect children and adults [47]. These comparisons can dramatize how fine particulate levels affect acute and overall health.

## 5. Conclusions

Our experimental investigation of levels of PM_2.5_ in cars adds to the known damage from outdoor ambient air pollution in Thailand. Our monitoring of PM_2.5_ was designed to conservatively reflect the real-world exposure of smoking one cigarette in a twenty-minute period with various air exchange and air ventilation circumstances tested. PM_2.5_, with the most common air exchange and ventilation circumstances, was three and nearly five times the WHO 24 h standard in the back and front seats of the ten vehicles tested, respectively. The long-term WHO standard for outdoor ambient exposures is lower, at 5 μg/m^3^, and also exceeded Thailand’s higher interim standard of 15 μg/m^3^ for long-term exposure. PM_2.5_ levels were at least double the outdoor ambient levels when vehicles were operated with the most common air ventilation condition, and only one cigarette was smoked. Our results suggest it is vital that restrictions on smoking in cars are mandated for the health of children and adults significantly affected by SHS. 

There is now universal recognition that environmental factors significantly impact our lives and that care for the air, water, and land must be ensured. Damage from air pollution, as measured by PM_2.5_, is now recognized as a significant threat to global health. How people understand and take action to address smoke in enclosed places like cars will have significant future health consequences.

## Figures and Tables

**Figure 1 ijerph-20-06970-f001:**
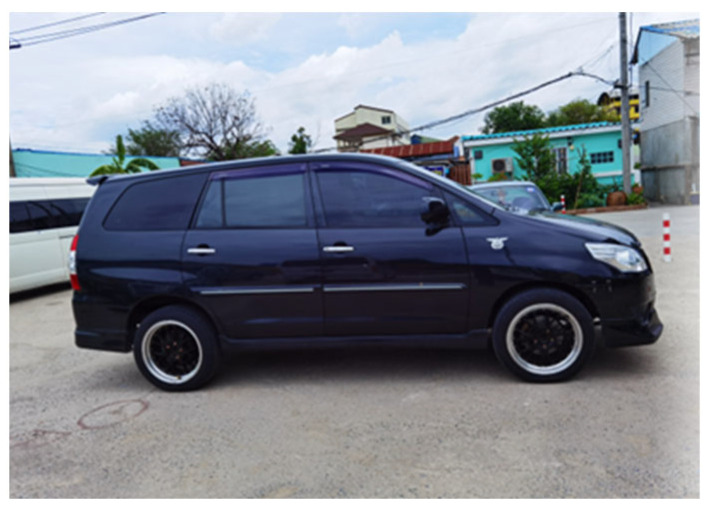
Car.

**Figure 2 ijerph-20-06970-f002:**
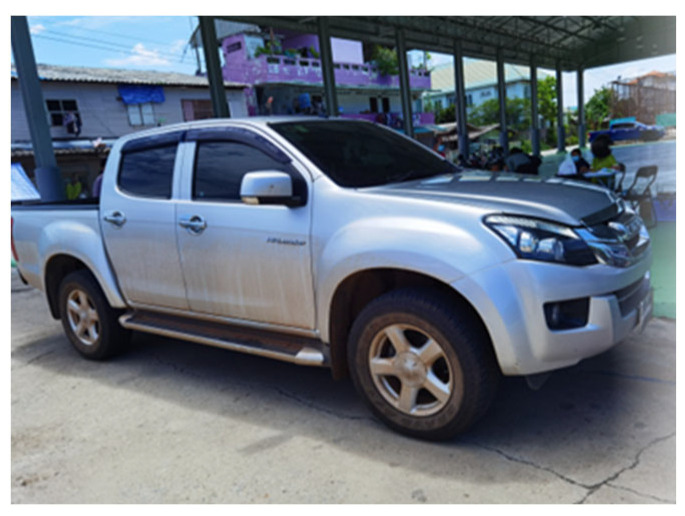
Truck.

**Figure 3 ijerph-20-06970-f003:**
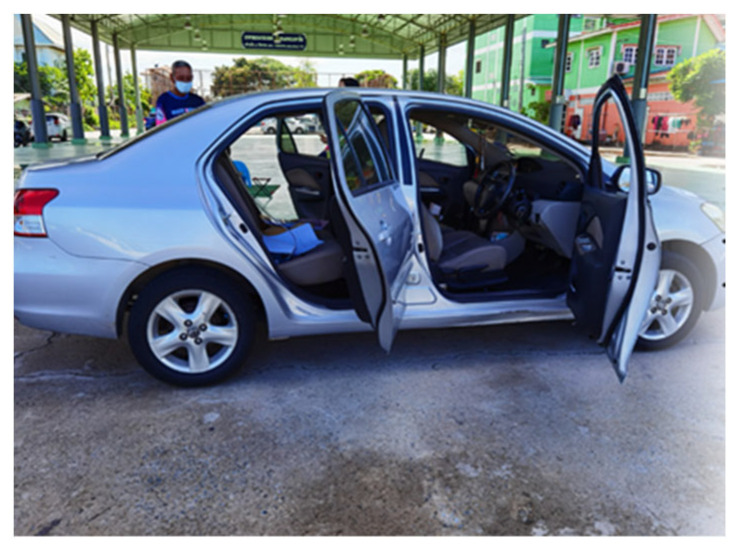
The car doors and windows were opened for at least 5 min between each experimental condition.

**Figure 4 ijerph-20-06970-f004:**
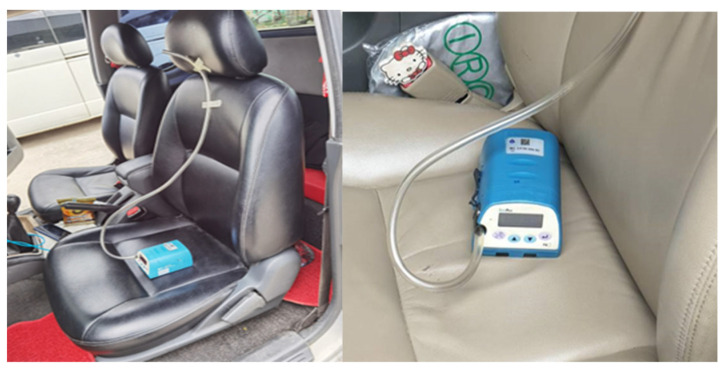
The placement of monitoring instruments in the front and back seats of each vehicle.

**Table 1 ijerph-20-06970-t001:** WHO-recommended AQG levels and interim targets.

Pollutant	Averaging Time	Interim Target				AQG Level
		1	2	3	4	
PM_2.5_, μg/m^3^	Annual	35	25	15	10	5
	24 h ^a^	75	50	37.5	25	15
PM_10_, μg/m^3^	Annual	70	50	30	20	15
	24 h ^a^	150	100	75	50	45
O_3_, μg/m^3^	Peak season ^b^	100	70	-	-	60
	8 h ^a^	160	120	-	-	100
NO_2_, μg/m^3^	Annual	40	30	20	-	10
	24 h ^a^	120	50	-	-	25
SO_2_, μg/m^3^	24 h ^a^	125	50	-	-	40
CO, mg/m^3^	24 h ^a^	7	-	-	-	4

^a^ 99th percentile (i.e., 3–4 exceedance days per year). ^b^ Average of daily maximum 8 h mean O_3_ concentration in the six consecutive months with the highest six-month running-average O_3_ concentration. Source: https://cdn.who.int/media/images/default-source/air-pollution/recommended-aqg-levels-and-interim-targets.jpg?sfvrsn=9f9136ef_2, accessed on 15 August 2023.

**Table 2 ijerph-20-06970-t002:** Total mean PM_2.5_ level with ambient and increase from smoking in the “front seat” of ten monitored vehicles. Values are the mean ambient PM_2.5_ plus the PM_2.5_ generated by smoking.

Condition	Mean PM_2.5_ Total (Ambient Plus Smoke Increase), Rounded to Whole Number		24 h PM_2.5_ Standard	
			WHO (15 μg/m^3^)	Thailand Interim (37.5 μg/m^3^)
1. Moving Open	(13)	30 μg/m^3^	✕	√
2. Parked Open	(30)	47 μg/m^3^	✕	✕
3. Moving Closed *	(55)	72 μg/m^3^	✕	✕
4. Parked Closed	(148)	165 μg/m^3^	✕	✕

Note: * Condition 3 is the smokers’ preferred condition for ventilation while driving in Thailand. Values in parentheses ( ) are PM_2.5_ mean increases from smoking in the vehicles (less the outdoor, ambient mean levels in the experimental settings, which were 17.1 μg/m^3^). √ Not over 15 μg/m^3^, the 24 h WHO standard. ✕ means exceeds the WHO or Thailand 24 h standard.

**Table 3 ijerph-20-06970-t003:** Total mean PM_2.5_ level with ambient and increase from smoking in the “back seat” of ten monitored vehicles. Values are the mean ambient PM_2.5_ plus the PM_2.5_ generated by smoking.

Condition	Mean PM_2.5_ Total (Ambient Plus Smoke Increase), Rounded to Whole Number		24 h PM_2.5_ Standard	
			WHO (15 μg/m^3^)	Thailand Interim (37.5 μg/m^3^)
1. Moving Open	(5)	22 μg/m^3^	✕	√
2. Parked Open	(17)	34 μg/m^3^	✕	√
3. Moving Closed *	(32)	49 μg/m^3^	✕	✕
4. Parked Closed	(107)	124 μg/m^3^	✕	✕

Note: * Condition 3 is the smokers’ preferred condition for ventilation while driving in Thailand. Values in parentheses ( ) are PM_2.5_ mean increases from smoking in the vehicles (less the outdoor, ambient mean levels in the experimental settings, which were 17.1 μg/m^3^). √ Not over 15 μg/m^3^, the 24 h WHO standard. ✕ means exceeds the WHO or Thailand 24 h standard.

## Data Availability

Data of this study has been reported to the THPI and Thai Health Promotion Foundation, but there is no public accessible. Requests to the THPI or its authors is necessary for data access.

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
