# Peer review of "Air Pollution inside Vehicles: Making a Bad Situation Worse"

_ijerph, 2023, doi:10.3390/ijerph20216970_

Round 1

Reviewer 1 Report

Review

General impression

This experimental study investigates inside-vehicle PM2.5 concentrations due to smoking inside private cars. In confined spaces, concentrations can reach very high levels and can pose a health risk. A relevant and important topic to ask for attention for. Particularly in view of exposure to secondhand tobacco smoke through passive smoking of children. The paper is well written. And relevant literature cited. The conclusions seem however quite strongly formulated, which may not be justified (based on these data) methodologically. For example because authors compare measured short term increases with Standards (WHO) for long term exposure (annual average concentrations). These concentration levels cannot be compared to claim exceedance of standards this way. This needs to be checked. Still, it is important to warn for health risks of passive smoking.

Originality

The topic is interesting, and contributes to the existing evidence, though its novelty is modest. A large number of existing studies have investigated exposure in cars and several reviews have been published, including the review by Raoof et al (2015) which the authors refer to. The authors may elaborate more on the added value of their study.

Strengths

The measurements were taken in Bangkok, where background concentrations of PM2.5 might be high, and vary in time. The authors start their measurement with a baseline concentration assessment, to assess how the concentrations increase by smoking. Several conditions are tested with repeated measurement.

Limitations

Exposure of the smoker is higher than exposure through elevated levels inside the vehicle, therefore these results are mainly of interest in view of unvoluntary passive smoking, particularly when children are exposed, or other vulnerable people (e.g. people with pre-existing cardiovascular conditions, or respiratory disease).

Short-term peak levels cannot be compared with WHO limit levels for an annual average concentration.

The amount of sigarettes smoked and the brand of the sigarettes were not standarised. Some smokers in the experiment smoked more than just 1 sigarette during the measurement interval of 20 minutes, and smokers smoked different brands. It is not clear how this may have affected the results.

What it aims to add to the subject area

The study aims to provide insight in inside-vehicle exposure to PM2.5 while smoking in private cars. This is of interest to assess the phenomenon of passive smoking, where passengers (sometimes children) are unvoluntary exposed to tobacco smoke.

The topic is interesting, and contributes to the existing evidence, though its novelty is modest. A large number of existing studies have investigated exposure in cars and several reviews have been published. However this study adds to the knowledge base with new data, for the current situation, and representative inside-car conditions in Thailand.  

Concerns and recommendations

11)      What the study aims to contribute:

The authors may more thoroughly consider the discussion of the relevance and implications of their study. e.g.:

The topic is interesting, and contributes to the existing evidence, though its novelty is modest. The focus on this aim of the study: “introduce evidence for regulating smoking in cars.” may be reformulated. Since “introduce evidence” suggests there is no current evidence or data. While previous studies have been published on the topic before (and have reported much higher inside-vehicle concentrations). A large number of existing studies have investigated exposure in cars and several reviews have been published, as the authors refer to.

These data add to the body of evidence, and support warning purposes (to warn people not to expose children or their non-smoking passengers).

Does the study provide support for regulations of public transport vehicles such as taxi/cab? Does that seem a more logical option: bans tot prevent exposure in public spaces in view of unvoluntary exposure of visitors, as compared to private spaces? Bans or regulations for private places, like the home or a persons own car are subject to much more controverse. The authors may explain more on how this was approached for passive smoking in homes.

Part of the added value of this study lies also in the fact that it contributes to assessment of population exposures in Thailand, and thereby can contribute future Health Impact Assessments (HIA) to assess the impact on public health.   

22)      Measured levels and conclusions drawn:

Peak exposure levels rose an average of 56 and 180 μg/m3 in the front seat and 29 and 75 μg/m3 in 24 the back seat.”

These levels seem comparatively low as compared to measurements reported from previous studies. Much higher in-vehicle concentrations have been reported previously. See e.g. a study by Schober et al 2019 (Passive exposure to pollutants from conventional cigarettes and new electronic smoking devices (IQOS, e-cigarette) in passenger cars.) who found PM2.5 values ranging from 64 up to as high as 1988 µg/m³ (!) while smoking tobacco cigarettes in cars. Or the review by Raoof et al., 2015. (A systematic review of secondhand smoke exposure in a car: Chronic Respir. Dis. 2015;12:120–131), who report that “Among all studies that assessed smoking in cars with at least one window partially open, the particulate matter 2.5 μm or less in diameter (PM2.5) concentrations ranged from 47 μg/m(3) to 12,150 μg/m(3). For studies with all windows closed, PM2.5 ranged from 203.6 μg/m(3) to 13,150 μg/m(3)”.

Even though previous papers have reported inside vehicle concentrations as high as 12150 ug/m3, this current paper concludes quite strongly: “These PM2.5 exposure levels in vehicles push vehicle exposures above current WHO and Thai standards to dangerous levels, necessitating regulatory action to limit vehicle smoking”. Is this conclusion justified? It is good to warn for the health risks. At the same time, it might need to be checked if the results justify such strong conclusions and it may therefore be formulated more carefully? The authors may in the discussion, reflect on differences between the results of their study and previous research. And may discuss why the concentrations found in their study seem lower than concentrations reported in previous studies.

Other environments with high PM2.5 concentrations include tunnels. Reported concentrations are in the same order of magnitude as concentrations reported here. A review showed an average PM2.5 concentration of 107 μg/m3 (See Marinello et al, 2020. Roadway tunnels: A critical review of air pollutant concentrations and vehicular emissions).  With higher levels up to 398 µg/m3 reported for a tunnel in China.  See: C. Huang, S. Tao, S. Lou, Q. Hu, H. Wang, Q. Wang, L. Li, H. Wang, J. Liu, Quan, L. Zhou. Evaluation of emission factors for light-duty gasoline vehicles based on chassis dynamometer and tunnel studies in Shanghai, China. Atmos. Environ., 169 (2017), pp. 193-203,

Minor textual:

Please check sentence 326: “We focused on There is a renewed call by researchers for action and public awareness and sentiment to address the growing respiratory and cardiovascular health consequences.”

English is fine. Only minor textual corrections and checking required. 

Reviewer 2 Report

INTRODUCTION

The authors exposed the timeline of similar studies and explained the impact caused by laws regarding banning smoke from public places. I would love to see a paragraph highlighting the effects of smoking on people and another one regarding the synergy between smoking and air pollution levels since the authors cited it in the introduction.

MATERIALS AND METHODS

The methodology is well written, but I would like to see a picture of the sampler, a picture of the used vehicles, a picture of the samples installed in the vehicles, and a map of the route. Or the route was random, and the authors only accounted for the time? I would like to see the calibration methods and calibration certificate in the supplementary material.

I would like to see the dates and hours the experiments were conducted and how many cigarettes each driver smoked during it. This will help other studies to replicate your conditions, and it also helps to explain the concentration among the tests. Additionally, showing the time series of the concentrations will help to identify the smoke peak across the experiment.

RESULTS

The authors refer to the concentration as "increase above ambient outdoor". This is a little confusing. Does this mean that the author subtracted the observed value by the ambient concentration, or the exposed values in the tables are the observed concentrations inside the car? Please clarify.

Why do the authors only compare with the annual air quality standard? The exposure in this test is acute, so I would compare it with short-term air quality standards. If the authors want to discuss the long-term exposure, they need to assume some conditions for the drivers to extrapolate to the log-term, such as assuming the driver is using the car every day under the same conditions. 

I recommend showing the time series collected in the samplers since it collects data 60 by 60 seconds. This will show how the concentrations vary during the experiment.

DISCUSSION

The discussion focused in the compare authors' findings with another study, but the other studies found higher concentrations. The authors need to explain why.

In the introduction, the authors mention they will compare the results with ambient air quality, especially over the "burning" season, but I did not see it in the paper. It would be nice to see the PM2,5 concentration for the experiment day and a comparison between the two. This will give a baseline for comparison and show how smoking will increase exposure to PM2,5, since the authors brought some papers that confirm it.

CONCLUSION

The conclusion needs work. The authors state common sense, such as we need laws to improve the air quality, but they need to show the need in the discussions. Expose your results, how high are the concentrations, hoe they vary, compare with short and long-term air quality standards, compare with ambient air quality to expose how high are the concentrations inside the vehicles, and then you can have a great conclusion

The paper is well written with good English and it is easy to read

Round 2

Reviewer 2 Report

Dear Authors,

Thank you for addressing the comments.